# Time-Aware Learning Framework for Over-The-Top Consumer Classification Based on Machine- and Deep-Learning Capabilities

**Jaeun Choi** [1] and **Yongsung Kim** [2,*]

1   Department of Artificial Intelligence Software, Kyungil University, Gyungbuk 38428, Korea; juchoi@kiu.kr
2   Department of Software Engineering, The Cyber University of Korea, Seoul 03051, Korea
*   Correspondence: kys1001@cuk.edu; Tel.: +82-2-6361-1948

**Abstract:** With the widespread use of over-the-top (OTT) media, such as YouTube and Netflix, network markets are changing and innovating rapidly, making it essential for network providers to quickly and efficiently analyze OTT traffic with respect to pricing plans and infrastructure investments. This study proposes a time-aware deep-learning method of analyzing OTT traffic to classify users for this purpose. With traditional deep learning, classification accuracy can be improved over conventional methods, but it takes a considerable amount of time. Therefore, we propose a novel framework to better exploit accuracy, which is the strength of deep learning, while dramatically reducing classification time. This framework uses a two-step classification process. Because only ambiguous data need to be subjected to deep-learning classification, vast numbers of unambiguous data can be filtered out. This reduces the workload and ensures higher accuracy. The resultant method provides a simple method for customizing pricing plans and load balancing by classifying OTT users more accurately.

**Keywords:** consumer classification; deep learning; machine learning; over-the-top; time-aware classification

## 1. Introduction

With the advancements of smart devices and the rapid development of wired and wireless networks, our modes of entertainment and venues for information are changing rapidly. In the past, our receipt of multimedia mainly relied upon standardized broadcasts from franchise television (TV) networks. With the turn of the century, our multimedia consumption has centered around internet smartphones, as typified by the "over-the-top" internet-protocol (IP) services of YouTube and Netflix [1]. Because of its ubiquity and simplicity, OTT content can be viewed worldwide without TVs. According to PricewaterhouseCoopers, the global OTT market is expected to grow sharply from USD 81.6 B in 2019 to USD 156.9 B in 2024 with an annual average growth of approximately 14% [2]. Furthermore, subscription OTT services are expected to reach approximately 650 million by 2021 [3]. With this growth, the dominant OTT players such as YouTube, Netflix, Amazon Prime, and Hulu, are actively promoting entry into the global market, and Disney, Apple, Warner Media, and HBO are gaining access using their existing content. The rising OTT market is indeed becoming a fierce battleground.

Consequently, network broadcast media and more recent TV-over-IP enterprises have experienced heavy competition [4,5]. On the other hand, internet-service providers (ISP) increasingly find themselves in conflicts with OTT media providers because of bandwidth- and fee-related issues. Because OTT services consume a vast amount of network resources, ISPs seek to bolster their profits to support growth [6]. Furthermore, consumers increasingly demand higher quality of service (QoS) from their ISPs [7] who desperately need to expand their throughput capabilities [8]. Often, the ISPs respond to

bandwidth overloads by limiting the amount of throughput (i.e., throttling) based on the OTT service in demand. This process is most often reactionary, inevitably creating surges of consumer complaints. To balance this vicious cycle of competing demands, the ISPs require better and more-timely consumer OTT-usage analysis capabilities, so that they can better mitigate network performance issues while balancing customer demand. Furthermore, utilizing a sound and trustworthy tool such for this purpose would put the ISPs in a better position to negotiate with OTT providers [9].

Traffic analysis models have been widely researched and utilized for this purpose in conventional hypertext-transfer-protocol (HTTP) mobile-network environments. Machine learning has been key to their success. However, only a few academic studies have focused on OTT content in the context of strategic service provision [9]. In this study, we analyze network consumption patterns based on consumers' OTT-usage patterns, confirming that a combination of machine- and deep-learning capabilities can achieve the highest accuracy. Additionally, by mitigating the time and resource requirements of deep learning, we provide a novel MetaCost-based framework related to OTT user analysis that can reduce the time required for analysis while exploiting the technology's high accuracy. This framework drastically reduces the analysis workload, making the process very efficient and timely so that ISPs can achieve instantaneous status and influence over OTT service demands.

This paper is structured as follows. In Section 2, we examine why the analysis of OTT-related trends and data-usage patterns is critical. We also justify the application of machine and deep learning to this pursuit with a review of previous traffic-analysis studies. In Section 3, we fully describe the OTT user-analysis framework. Section 4 presents a discussion of our experimental results. Finally, Section 5 presents the conclusions of this study with further research directions.

## 2. Literature Review

### 2.1. OTT Services

The success of OTT services is owed, in part, to the increase in the number of single-person households and their desire for highly personalized content. With the phenomena of cord-cutting, which circumvents paid broadcasting, and cord-shaving, which leverages alternate broadcasting venues, the demand for new and innovative OTT services will not likely decrease [10]. For many worldwide consumers, OTT services have replaced legacy subscription models. In Korea and China, consumers spend around USD 3 per month for high-definition OTT services with tailored recommendation systems [11].

Currently, the OTT market is dominated by giant companies such as Netflix, YouTube, Amazon Prime Video, and Hulu, which account for approximately 75, 55, 44, and 32% of the US OTT market, respectively. As a whole, these firms currently account for 79% of the US market share [12]. Competitors are quickly entering the fray. Disney launched Disney+ in November 2019 after acquiring 21st Century Fox, securing 50-million US subscribers as of July 2020 [13]. After acquiring Warner Media, AT&T launched HBO Max, which utilizes current and past HBO content, securing more than 34-million US subscribers as of June 2020 [14].

The rapidly changing OTT landscape presents both a crisis and an opportunity for conventional TV networks. Although it proves to be a disadvantage to extant strategies, it does provide an opportunity to enter new markets by providing OTT services using their current content and service supply chains [5]. Hulu, launched in 2008, dominates the market with content from FOX, NBC, and ABC [4]. In Korea, there are ~3-million monthly Wavve subscribers, which offers content from KBS, MBC, and SBS [15].

The market positions and strategies of ISPs are more complicated than those of legacy TV networks. ISPs that simultaneously provide IP-TV and internet services must install and maintain high-quality broadband infrastructures for both. Additionally, bundling strategies are required to prevent cord-cutting that would cancel IP-TV services in favor of OTT services alone [16]. In fact, in Korea, KT, which holds the highest share of the IP-TV market, is considering a strategy to create

synergy through a partnership with Netflix. As such, Netflix could use the opportunity to expand their market further into Korea. KT has the largest number of wired network subscribers and can collect more subscribers and their abundant network usage fees by providing Netflix content [17]. With the gradual distribution of 5G, the number of customers using wireless OTT services is bound to grow. If stable QoS cannot be guaranteed, customer churn will be difficult to deal with. In particular, with the increase of the use of real-time video-streaming services (e.g., Twitch and Discord) stable services will forever be challenged [8,18].

As mentioned, ISPs must be able to dynamically execute service degradation plans to minimize network-resource overconsumption while meeting the QoS needs of consumers. That is, it is essential to create a win–win situation for both ISPs and OTT providers. Based on assumptions of network neutrality, various studies have analyzed the complex pricing systems related to content providers, networks, and consumers. Dai et al. [7] proposed a pricing plan that could guarantee QoS based on the Nash equilibrium. They showed that the direct sale of QoS by an ISP to a consumer achieved better results than selling QoS to an OTT provider. Based on the quality of experience (QoE), a model that would benefit all OTT providers, networks, and consumers was also proposed [19]. This study compared three methods: Providing better QoE to customers that paid more; satisfying QoEs of the most profitable customers (MPC) to increase lifetime value; and providing fair QoE to all customers. Of these, the method of providing QoE to MPCs was found to be the most beneficial. A study based on shadow pricing was also conducted to determine an effective method to price broadband services [20], concluding that setting the pricing plan according to the usage patterns of consumers was the best strategy. Because OTT services utilize a considerable amount of network bandwidth, some studies have proposed methods of predicting network consumption and pricing via a content-delivery or a software-defined network [21,22]. With the spread of OTT services, the pricing-related issues of OTT and network providers persist, and most existing studies have suggested plans based on the amount of network usage. Thus, in order for networks or OTT providers to establish an optimal strategy related to OTT, the OTT-service usage patterns of consumers must be identified very quickly. Both network and OTT providers can establish effective pricing strategies only when they can correctly and immediately identify which users are using what OTT services and how much data they consume, classifying all items and load-balancing accordingly. Thus, OTT user classification is the first step in establishing an effective strategy.

*2.2. Review of Classification Using Machine Learning*

Researchers have conducted extensive studies on methods to manage and operate networks by analyzing network traffic and user behaviors. Hence, the widespread use of network technologies incorporating artificial-intelligence (AI) technologies has gained attention. Extensive research has been conducted on the knowledge-defined-networking paradigm, in which AI technology is incorporated into network routing, resource management, log analysis, and planning. Several companies have already applied AI data analysis to network operations [23]. In turn, multiple studies have been conducted to determine how network providers can leverage machine learning to analyze user traffic. Middleton and Modafferi [24] exploited machine learning to classify IP traffic in support of QoS guarantees. Yang et al. [25] proposed the classification of Chinese mobile internet users into heavy and high-mobility users by analyzing the network traffic of 2G and 3G services. Various other studies proposed methods, such as decision trees [26,27], support-vector machine (SVM) [28–30], *k*-nearest neighbor (KNN) [31,32], hidden Markov model (HMM) [33,34], and K-Means [35,36] for traffic classification. The application targets of these methods differed depending on whether the analysis was performed on wired, wireless, or encrypted traffic. However, these techniques demonstrated the following structure: They captured and analyzed network traffic; they applied machine learning by using traffic characteristics as features; and they classified the traffic data. The traffic data were diverse, ranging from captured packets to public datasets. However, data related to OTT usage, which is a recent trend, were rarely considered. Few studies have proposed methods to classify users based

on OTT consumption [9,37]. Those that did classified consumers into three consumption categories (i.e., high, medium, and low) by using various machine-learning methods to analyze OTT traffic. The current study is significant in that it is the first to use deep learning in an attempt to classify users in terms of OTT usage. However, deep learning has the disadvantage of requiring large numbers of calculations. On the other hand, it has the advantage of high accuracy. Hence, it has been widely utilized for similar classification problems [38–40]. Therefore, in this study, to overcome the demerits of excessive time-consumption, we propose a time-aware user-analysis framework that applies the MetaCost method [41]. Based on Bayes risk theory, MetaCost can reduce specific classification errors while setting the cost of misclassification differently. By using these properties, we can reduce the load on deep learning through cost adjustment.

## 3. Research Design

First, we verified whether OTT users can be effectively classified using machine- and deep-learning methods. The description and application method of the machine- and deep- learning used in this study are detailed in Section 3.1. In addition, we were able to confirm that the classification accuracy was high when using deep learning; however, it was also found that the time required was large due to the characteristics of deep learning. Therefore, in Section 3.2, we propose a framework that can reduce time consumption but utilizes the accuracy of deep learning as well.

### 3.1. Appling Machine- and Deep- Learning to OTT Consumer Classification

Figure 1 shows the process of analyzing OTT users based on machine- and deep- learning [42,43]. The steps include raw data collection, data preprocessing for feature extraction, and dataset processing for machine learning. We leverage OTT usage data for this purpose. Our dataset was previously published [37] and is open to the public. It includes general network traffic characteristics but is also appropriate for OTT-specific research. It contains traffic information about the activities of actual internet users with respect to 29 types of OTT services. A detailed description of the dataset is presented in Section 3.3.

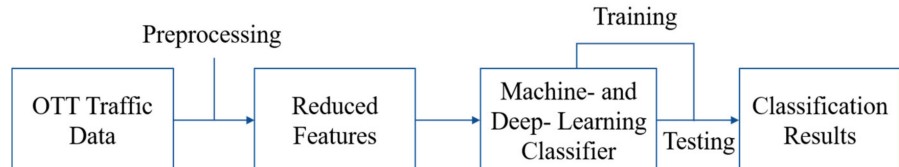

**Figure 1.** Machine- and deep-learning-based over-the-top (OTT) consumer classification process.

In this study, we analyze OTT users according to the conventional machine-learning methods of KNN, decision tree, SVM, naïve Bayes, and repeated incremental pruning to produce error reduction (RIPPER) models. We also use the multilayer perceptron (MLP) and convolutional neural network (CNN) as deep-learning applications.

### 3.1.1. Conventional Machine Learning Methods

**The KNN** is a typical classification method that applies clustering. It first confirms which class the $k$ neighbors of a data point belong to, and it then performs classification by taking the majority vote based on the result. If $k = 1$, it is assigned to the nearest-neighbor class. Therefore, if $k$ represents an odd number, classification becomes easier, because there is no possibility of a tie [44]. A neighbor in KNN can be defined by calculating the distance between vectors in a multidimensional feature set. The distance between vectors is calculated using Euclidean and Manhattan distances. Suppose that

input-sample $x$ has $m$ features. The feature set of $x$ is expressed as $(x_1, x_2, \cdots, x_m)$, and the Euclidean and Manhattan distances of $x$ and $y$ are defined as follows [45]:

$$\text{Euclidean}: D(x,y) = \sqrt{\sum_{i=1}^{m}\left|x_i - y_i\right|^2}, \tag{1}$$

$$\text{Manhattan}: D(x,y) = \sum_{i=1}^{m}\left|x_i - y_i\right|. \tag{2}$$

The greatest advantage of the KNN is its straightforwardness, and the variables are not required to be adjusted. This method only requires the assignment of a $k$ value. However, if the distribution is distorted, KNN cannot be applied, because the data may not belong to the same class, even if it is close to its neighbors. Additionally, when dealing with multidimensional data having many features, classification accuracy may be degraded, owing to the curse of dimensionality. Thus, it is essential to reduce features [46].

**Decision trees** are built upon the tree model and re-used to classify an entire dataset into several subgroups. When traversing from an upper to a lower node, nodes are split according to the classification variables. Furthermore, nodes of the same branch have similar attributes, whereas nodes of different branches have different attributes. The most typical decision-tree algorithms are ID3 and C4.5. The C4.5 algorithm, derived from ID3, minimizes the entropy sum of the subsets by leveraging the concept of "information gain". The subset is split to the direction that maximizes information gain. Thus, accuracy is high when classification is performed through the learned result. Decision trees have the advantages of intuitiveness, high classification accuracy, and simple implementation. Therefore, they are widely adopted for various classification tasks. However, for data containing variables at different levels, the level is biased mainly to most of the data. Moreover, for a small tree with fewer branches, rule extraction is easy and intuitive, but accuracy may decrease. Moreover, for a deep and wide tree having many branches, rule extraction is difficult and non-intuitive, but accuracy may be higher than that of a small tree.

**The SVM** performs classification based on a hyperplane that separates two classes in the feature space. The hyperplane having the longest distance between the closest data points to the hyperplane in two classes is set to have the maximal margin. For inputs $x$ and $y$, the hyperplane separating the classes is defined as $w^T \cdot x + b = 0$. After finding the distance between the hyperplane and closest data points of two classes, the optimization equation for maximization is defined as follows:

$$\min_{w,\,b} \Phi(w) = \frac{1}{2}\|w\|^2, \tag{3}$$

where variables $w$ and $b$, which satisfy the following convex quadratic programming, become variables that build the optimal hyperplane [45]:

$$\text{s.t } y_i\left(w^T x_i + b\right) \geq 1,\ i = 1, \cdots, l. \tag{4}$$

The SVM achieves high performance for a variety of problems. It is also known to be effective for the case of many features. Although SVM solves binary-class problems and can be applied to multiclass problems having various classes, it must solve multiple binary-class problems to derive accurate results. Thus, its calculation time is relatively long [45,46].

**The naïve Bayes method** is a typical classification method based on the statistical assumptions of the Bayes' theorem. It starts from the assumption that all input features are independent. When this is true, classes can be assigned through the following process [46]:

$$y(f_1, f_2, \cdots, f_m) = argmax_{k\in\{1,\cdots,K\}}p(C_k)\prod_{i=1}^{m}p(f_i|C_k), \tag{5}$$

where $m$ is the number of features, $k$ is the number of classes, $f_i$ is the $i$th feature, $C_k$ is the $k$th class, $p(C_k)$ is the prior probability for $C_k$, and $p(f_i|C_k)$ is the conditional probability for feature $f_i$ given class $C_k$.

The key merit of naïve Bayes is its short calculation time for learning. This is because, under the assumption that features are independent, high-dimensional density estimation is reduced to 1D kernel-density estimation. However, because the assumption that all features are independent is unrealistic, accuracy may decrease when performing classification using only a small sample of data. To increase accuracy, a large amount of data should be collected [46,47].

**The RIPPER** algorithm is a typical rule-set classification method [48]. Rules are derived by training the data using a separate-and-conquer algorithm. In turn, the rules are set up to cover as many datasets as possible, as developed using the current training data. The rules are pruned to maximize performance. Data correctly classified according to the rules are then removed from the training dataset [46]. The RIPPER algorithm overcomes the shortcoming of early rule algorithms, wherein big data could not be effectively processed. However, because the RIPPER algorithm starts by classifying two classes, performance can decrease when the number of classes increases. Its performance may also decrease because of its heuristic approach.

### 3.1.2. Deep Learning

After the AlphaGo (AI) beat Lee Se-dol (human) in the 2016 Google DeepMind Go challenge match, deep learning captured the attention of the worldwide public. However, research on deep learning had already been actively underway in academia and practical application fields. Deep learning is an extension of the artificial neural network, and it learns and makes decisions by configuring the number of layers that make up its neural network. It took a while for computer hardware to catch up, but with recent graphical processing-unit developments, deep learning has been widely applied in various fields. Deep learning automatically selects features through its training process. It does not require much assistance from domain experts and learns complex patterns of constantly evolving data [43]. As such, many related studies on internet traffic analysis have been published [49]. MLP and CNN deep-learning methods are used for this paper.

**The MLP** has a simple deep-learning structure and comprises an input layer, an output layer, and a hidden layer of neurons. Figure 2 shows the structure of the MLP. In each layer, several neurons are connected to the adjacent layer. The neurons calculate the weighted sum of inputs and output the results via a nonlinear activation function. In this process, the MLP uses a supervised back-propagation learning method. Because all nodes are connected within the MLP, each node in each layer has a specific weight, $w_{ij}$, with all nodes of the adjacent layer. Node weights are adjusted based on back-propagation, which minimizes the error of the overall result [49]. However, the MLP method has the disadvantage of being very complex and inefficient, owing to the huge number of variables the model must learn [43]. Accordingly, to use an MLP, it is necessary to acquire data that is not too complex or to pay close attention to time consumption. The OTT dataset used in this study has quantitative values for each feature. Since it does not have complicated feature structures such as images or videos, it shows sufficiently good performance even if only simple MLP is applied. Therefore, we tried to save the time required for learning and detection by using the simplest MLP structure possible. In this study, we conduct an experiment with an input layer, an output layer, and a single hidden layer between them.

**The CNN** is similar to the MLP, comprises several layers, and updates variables through learning. Although the MLP does not handle multi-dimensional inputs well, the CNN does so by applying a convolution layer. Figure 3 shows the structure of the CNN. The convolution layer produces results for the next layer by using kernels with learnable variables as inputs. The local filter is used to complete the mapping process, which is regarded as a convolution function. Additionally, because it is replicated in units, it shares the same weight vector and bias, thus increasing efficiency by greatly reducing the number of parameters. CNNs use a pooling process for down-sampling and can be widely applied to a variety of classifications. If the dimension of a vector used in the CNN process is 1, 2, or 3, it corresponds to 1D-, 2D-, and 3D-CNNs, respectively. The 1D-CNN is suitable for sequential data

(e.g., language), the 2D-CNN is suitable for images or audio, and the 3D-CNN is suitable for video or large-volume images. Although there has been no research on classifying OTT traffic data by CNN, a study analyzing general network traffic via CNNs utilized a 1D-CNN [50]. This is because traffic characteristics are sequential; therefore, 1D-CNNs are sufficient and multi-dimensional CNNS are not required. In this study as well, OTT traffic was analyzed using a 1D-CNN since OTT traffic is similar to traditional traffic data. In addition, the filtering and pooling processes were performed using two convolutional layers as the complexity of the dataset was not high.

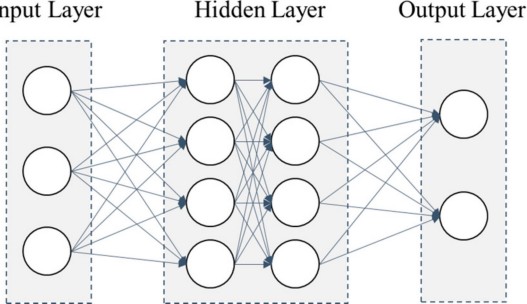

**Figure 2.** Multilayer perceptron (MLP) process.

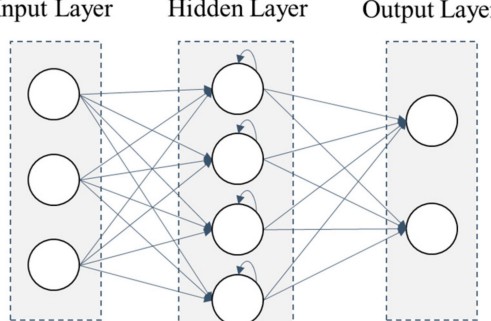

**Figure 3.** Convolutional neural network (CNN) process.

As described in detail in the experimental results of Section 4, when deep learning is applied to OTT user analysis, the accuracy is higher than when applying general machine-learning methods, although it takes much longer. The number of users of the dataset used in this study is 1,581, which is considerably less than the number of users serviced by ISPs or OTT providers. With larger numbers of simultaneous users, the time consumption could become prohibitive. Therefore, we apply the aforementioned MetaCost framework.

### 3.2. Time-Aware Consumer Classification Based on MetaCost and Deep Learning

In this study, we classify consumers into three consumption types (i.e., high, low, and average) by analyzing their OTT-usage traffic. Notably, there are two other classes of users that we ignore: Those that use an extremely heavy amount of OTT services and those who rarely use services. As these classes, which have extreme characteristics, can be easily classified via general machine learning, deep learning does not need to be used to classify these extrema. Therefore, we propose a framework that first filters high- and low-consumption consumers through a fast and relatively accurate machine-learning technique. Then, it performs deep-learning-based classification for the remaining customers. This framework shortens the overall computation time by reducing the number of samples to which deep learning is applied, allowing it to focus on the more ambiguous classes [41]. Figure 4 illustrates the proposed time-aware framework based on MetaCost.

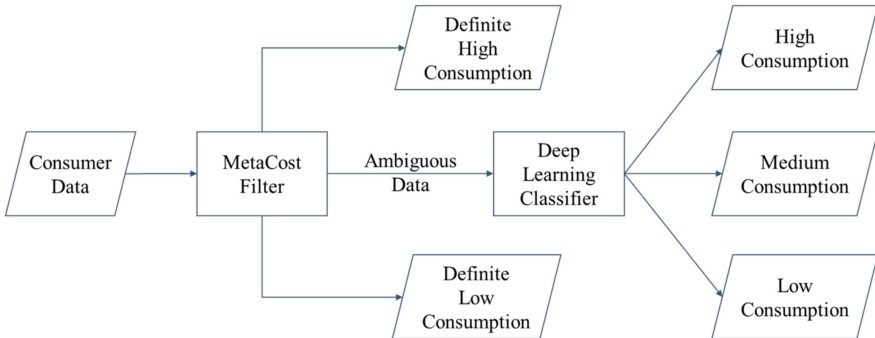

**Figure 4.** Proposed framework.

Figure 5 shows a simple schematic of the high- and low-consumption filtering process, which forwards the non-extrema data to the deep-learning-based classifier. By setting the cost of errors that misclassify non-high-consumption data as "high" greater than that of errors that incorrectly classify high-consumption data as "medium" or "low", we prevent other classes of data from being misclassified as "definite high-consumption". When classifying definite low-consumption data, we filter out only the obvious data by setting the cost with the contrapositive logic. If the cost is set high in order to not mix the filtered data with other data, all data that are slightly ambiguous are forwarded to the next step, as shown with the "high-cost" process of Figure 5. As a result, the load on deep learning is mitigated. However, there is an increased chance of lower accuracy, because, during the filtering process, medium-consumption data can be misclassified as high- and low-consumption data. We adjust the tradeoff relationship between accuracy and time-consumption by controlling costs according to the number of data and resource state.

In the case of general machine learning, the weights for the errors resulting from the classification process remain the same. The MetaCost method sets the cost of errors differently and is suitable to be applied to the proposed filtering framework, because classification is performed in terms of minimizing costs. The MetaCost method assigns each data to a class satisfying the following equation:

$$x's \text{ class} = \arg \min_i \sum_j p(j|x) \, C(i, j), \tag{6}$$

where $p(j|x)$ is the probability that $x$ belongs to class $j$, and $C(i, j)$ is the cost incurred when $x$ actually belongs to class $j$ but is classified as class $i$. After calculating the cost of misclassification for each datum, it is assigned to the class having the lowest cost [41].

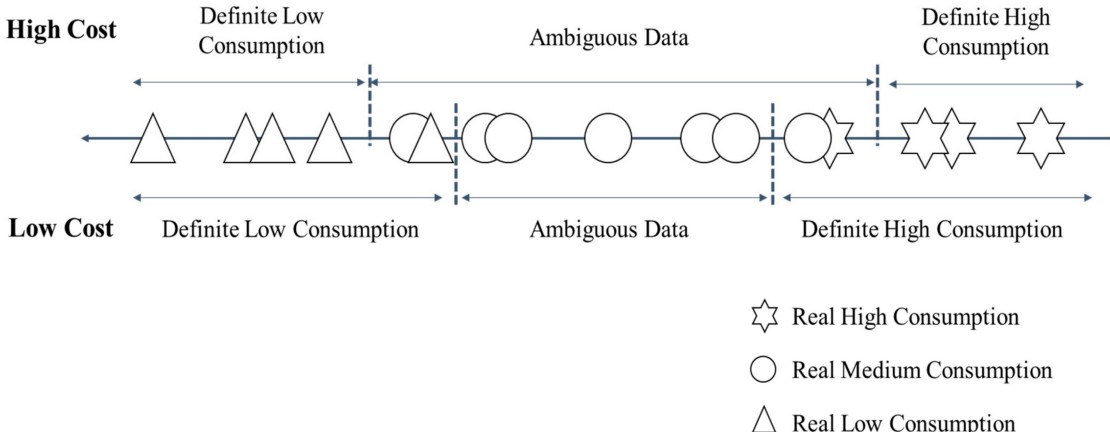

**Figure 5.** Filtering results according to cost.

The proposed framework's classification time is far less than that of the simple deep-learning method, but deviations can occur based on the cost setting. However, deep learning is generally applied after filtering more than half of the data, making it advantageous over the proposed framework in terms

of time consumption. Thus, even with greatly increased sizes of the analysis dataset, the strengths of proposed framework will stand out. As mentioned, the framework increases in flexibility via cost-setting adjustments. If the cost is properly adjusted according to the environment in which this framework is adopted, classification can be performed according to the time and accuracy desired by the analyst.

### 3.3. Dataset Description

To verify the proposed methodology, we applied the pre-existing dataset mentioned in Section 3.1. Traffic captured directly from the Universidaa del Caucau (Unicauca) network in 2017 was converted into a dataset comprising 130 features and 1581 user samples. These samples were divided into classes of high, medium, and low consumption. The OTT usage data were well represented. Twenty-nine applications were analyzed, including 29 OTT services: Amazon, Apple Store, Apple iCloud, Apple iTunes, Deezer, Dropbox, EasyTaxi, Ebay, Facebook, Gmail, Google suite, Google Maps, HTTP_Connect, HTTP_Download, HTTP_Proxy, Instagram, LastFM, MS OneDrive, Facebook Messenger, Netflix, Skype, Spotify, Teamspeak, Teamviewer, Twitch, Twitter, Waze, WhatsApp, Wikipedia, Yahoo, and YouTube. Features were extracted by analyzing the traffic flow of each service, as shown in Table 1. For each service, features were extrapolated from the dataset [37].

**Table 1.** Feature Description.

| Feature Name | Feature Description |
| --- | --- |
| Application-Name.Flows | Amount of the internet-protocol (IP) flow sent by individual users for each OTT service |
| Application-Name.Flow.Duration.Mean | Average time (s) spent by individual users for being connected to each OTT service |
| Application-Name.AVG.Packet.Size | Packet size sent by individual users for each OTT service |
| Application-Name.Flow.Bytes.Per.Sec | Byte size per second sent by individual users for each OTT service |

## 4. Results and Discussion

### 4.1. Machine and Deep Learning

For classification based on KNN, decision tree, SVM, naïve Bayes, and RIPPER, we employed Weka, a JAVA-based machine-learning library [51]. For MLP and CNN, we employed scikit-learn and TensorFlow. For the experiments, the hardware included an Intel i7-1065G7 processor, 16-GB LPDDR4x memory, and NVIDIA® GeForce® MX250 graphics with GDDR5 2-GB graphic memory. To select machine learning parameters with the best performance for each method, we experimented with adjusting the various parameters to find appropriate values for the OTT datasets. For the KNN, *k* was set to 17, and J48 was used as the decision tree. In the SVM, a linear kernel was used, and, in RIPPER, the number of folds used for pruning was set to 10. The naïve Bayes classifier used Weka's default settings. For the MLP, ActivationELU was used as the activation function of the hidden and output layers, ADAM was used as the optimizer of the loss function, and AdaDelta was used as the bias updater. For the CNN, two each of convolution, pooling, and fully connected layers were used. ActiviationIdentity and ActivationSoftmax were used as the activation functions of the convolution and output layers, respectively. Adamax was used as the optimizer of the loss function, and AdaDelta was used as the bias updater. Additionally, the number of epochs was set to 100.

We considered recall, precision, and F-measure as the evaluation metrics, calculating them based on the basic true positives (TP), false positives (FP), and false negatives (FN). Recall indicates the number of classes detected among the actual classes, and is the same as TP. Precision is the accuracy of detection and refers to the probability that, when a datapoint is classified into a class, it actually falls

into that class. F-Measure is used to obtain the harmonic average value for precision and recall and simultaneously indicates accuracy. These metrics are defined as follows:

$$\text{Recall} = \frac{TP}{TP+FN}, \text{Precision} = \frac{TP}{TP+FP},$$

$$\text{F} - \text{Measure} = \frac{2 \cdot \text{Recall} \cdot \text{Precision}}{\text{Recall} + \text{Precision}}.$$

(7)

Table 2 shows the experimental results based on the aforementioned environment. In the case of the conventional machine-learning methods, KNN achieved good performance with a classification accuracy of 95.1%, and SVM achieved a satisfactory performance of 92.9%. Except for naïve Bayes, all machine-learning methods showed detection rates over 90%, confirming their applicability in classifying OTT users. The accuracy of deep learning was even higher: The use of MLP and CNN to classify consumers achieved a detection rate of 98.2 and 97.6%, respectively. Because the data input was not complex, we observed that the accuracy of MLP was higher than that of CNN. Because both deep-learning methods achieved high performance, we confirmed that their application could also be effectively applied to OTT user analysis.

**Table 2.** Classification results of machine and deep learning. Abbreviations: k-nearest neighbor (KNN); support-vector machine (SVM); repeated incremental pruning to produce error reduction (RIPPER); multilayer perceptron (MLP); convolutional neural network (CNN).

| Machine Learning Algorithm | Recall | Precision | F-Measure | Time (s) |
|---|---|---|---|---|
| KNN | 0.951 | 0.951 | 0.950 | 0.3 |
| Decision tree-J48 | 0.918 | 0.918 | 0.918 | 0.6 |
| SVM | 0.929 | 0.928 | 0.928 | 48.6 |
| Naïve Bayes | 0.696 | 0.726 | 0.701 | 0.4 |
| RIPPER | 0.908 | 0.909 | 0.908 | 8.1 |
| MLP | 0.982 | 0.982 | 0.982 | 298.1 |
| CNN | 0.976 | 0.976 | 0.976 | 857.1 |

Tables 3 and 4 show the detailed classification results of the three types of consumers through deep learning. As shown in Table 3, MLP classified high- and low-consumption users with an accuracy of ≥98%. For medium consumption, although the classification accuracy was lower, it was relatively high at 96%. The results of the CNN shown in Table 4 show a similar tendency. The classification accuracy reached ~99% for high- and low-consumption users, whereas medium consumption showed a relatively accurate detection rate of 94.6%.

**Table 3.** Detailed classification results obtained through MLP.

| Real Data | Classified as | | | Recall | Precision | F-Measure |
|---|---|---|---|---|---|---|
| | High Consumption | Medium Consumption | Low Consumption | | | |
| High Consumption | **98.75%** | 1.09% | 0.16% | 0.988 | 0.980 | 0.984 |
| Medium Consumption | 2.60% | **96.31%** | 1.09% | 0.963 | 0.978 | 0.970 |
| Low Consumption | 0.21% | 0.63% | **99.16%** | 0.992 | 0.987 | 0.989 |

When classifying OTT users based on deep learning, the classification accuracy was observed to be relatively high, as in other applications fields. However, as confirmed by the classification times shown in Table 2, deep learning took longer to classify consumers than did conventional machine-learning methods. The next subsection describes the MetaCost savings.

**Table 4.** Detailed classification results of CNN.

| | Classified as | | | | | |
|---|---|---|---|---|---|---|
| Real Data | High Consumption | Medium Consumption | Low Consumption | Recall | Precision | F-Measure |
| High Consumption | **98.91%** | 1.09% | 0% | 0.989 | 0.974 | 0.981 |
| Medium Consumption | 3.46% | **94.60%** | 1.94% | 0.946 | 0.973 | 0.959 |
| Low Consumption | 0.21% | 1.05% | **98.74%** | 0.987 | 0.981 | 0.984 |

*4.2. Time-Aware Consumer Classification*

To reduce the time required for deep learning, we first classified high- and low-consumption data by using machine learning and MetaCost. We then classified only the remaining ambiguous data using deep learning. KNN and J48 decision trees were the machine learning methods used as the primary filter. Although KNN showed the best performance among all machine-learning methods, J48 achieved fast and highly accurate results. The cost requirement of applying MetaCost is defined as "the cost incurred when classifying data other than high/low consumption as high or low consumption". Therefore, with the cost set to "high", ambiguous data are forwarded to the secondary deep-learning classification. This experiment was performed while changing the cost from 1 to 30 in steps of five. If the cost was one, the weight for all errors was one. Accordingly, the result obtained was the same as that without the application of MetaCost. If the cost was set higher than one, classification was performed to reduce costs. At each step as the cost approached 30, no significant difference was observed from the previous step. Thus, to observe the most conspicuous difference, the experiment was conducted with the cost set to 30. In the secondary classification process, the MLP algorithm was applied for deep learning. Table 5 shows the processing results of the primary filter using KNN and J48 while adjusting the cost from 1 to 30. The table presents the number of data filtered by the primary filter, incorrectly classified by the filter, and processed by deep learning (the secondary classification) with the final detection time.

**Table 5.** Filtering result according to cost change.

| Cost | Number of Data First Filtered | | Number of Data Incorrectly Filtered | | Number of Data Processed by Deep Learning | | Time Taken for Detection (s) | |
|---|---|---|---|---|---|---|---|---|
| | KNN | J48 | KNN | J48 | KNN | J48 | KNN | J48 |
| 1 | 1167 | 1117 | 89 | 65 | 414 | 464 | 71.6 | 68.7 |
| 5 | 924 | 1009 | 14 | 27 | 657 | 572 | 111.6 | 88.3 |
| 10 | 792 | 965 | 5 | 13 | 789 | 616 | 145.2 | 99.5 |
| 15 | 719 | 950 | 2 | 7 | 862 | 631 | 146.9 | 102.2 |
| 20 | 668 | 949 | 2 | 6 | 913 | 632 | 173.9 | 102.0 |
| 25 | 633 | 953 | 2 | 19 | 948 | 628 | 177.5 | 101.8 |
| 30 | 613 | 958 | 2 | 21 | 968 | 623 | 178.7 | 101.6 |

With an increase in the cost setting, only the more obvious data were filtered out. Thus, the number of filtered data decreased, and those processed through deep learning increased, resulting in an increase in detection time. However, even if the cost was set to an extremely high value of 30, the detection time was about half. This resulted in the best accuracy while reducing the detection time by more than half. If the cost was set to "low", the detection time was reduced to approximately 23%. However, in this case, the number of incorrect classifications increased, negatively affecting the classification accuracy of the entire framework. When using KNN as the primary filter, results showed fewer errors. However, the number of filtered data was less than that when using J48. Therefore, KNN was determined to utilize more time than J48. On the contrary, although J48 utilized less time because of more filtering, it resulted in more errors. Therefore, the overall classification result of J48 was poor. Table 6 summarizes the overall classification accuracy of the framework per filtering method. Because of space limitations, the detailed results are included in the Appendix A. As shown in the

results of Table 6, with an increase in the cost setting, the ambiguous data were forwarded for accurate deep learning, leading to higher accuracy. In terms of accuracy, the results showed only a slight difference when classifying the entire dataset using deep learning. The filter using KNN showed higher accuracy, because it filtered less data than did the filter using J48, resulting in more data being processed during the classification step. Therefore, as observed, the use of the filter with KNN utilized more classification time than that did that of J48. Overall, the filters using KNN and J48 showed classification accuracies of 97 and 96%, respectively, with no significant difference from the value obtained using only deep learning. For both filters, with the cost set higher, the accuracy increased, but the classification time also increased, as shown in Figure 6. Overall, while the accuracy of KNN was high, it utilized more time. When analyzing OTT users, if a considerable amount of data must be analyzed, the focus should be on reducing the time by setting the cost low. Furthermore, if the data to be analyzed are relatively small or if there is sufficient time for analysis, accuracy can be improved by setting the cost high. Thus, optimal time and detection rates can be set while adjusting the cost according to the given environment.

**Table 6.** Overall classification accuracy of the framework according to cost change.

| Cost | KNN Filter + Deep Learning | | | J48 Filter + Deep Learning | | |
|------|------|------|------|------|------|------|
| | Recall | Precision | F-Measure | Recall | Precision | F-Measure |
| 1 | 0.930 | 0.936 | 0.927 | 0.945 | 0.947 | 0.944 |
| 5 | 0.968 | 0.968 | 0.968 | 0.963 | 0.963 | 0.962 |
| 10 | 0.969 | 0.969 | 0.969 | 0.967 | 0.967 | 0.967 |
| 15 | 0.972 | 0.971 | 0.971 | 0.967 | 0.967 | 0.967 |
| 20 | 0.973 | 0.973 | 0.973 | 0.968 | 0.968 | 0.968 |
| 25 | 0.974 | 0.974 | 0.974 | 0.968 | 0.969 | 0.968 |
| 30 | 0.976 | 0.976 | 0.976 | 0.968 | 0.969 | 0.968 |

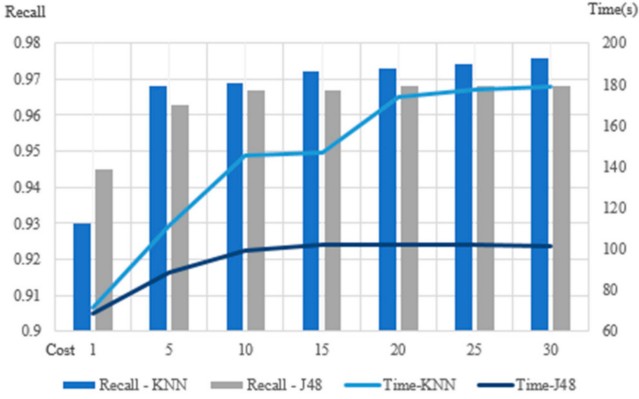

**Figure 6.** Changes in accuracy and time according to changes in the cost.

The dataset used in this study contained the metadata of 1581 people. Therefore, regardless of analysis time, a significant time difference was not observed. As mentioned, ISPs or OTT providers will likely face hundreds of thousands or millions of users. If the data corresponding to the actual number of users are analyzed using the proposed method, time savings will be clearly observed. To this end, by applying SMOTE [52], an oversampling method, we created a dataset with 159,681 instances. SMOTE is a technique for creating new samples based on existing samples. Unlike other oversampling techniques that simply duplicate existing samples, SMOTE creates synthetic data based on existing data. Therefore, it is possible to create a dataset that has similar characteristics to an existing dataset but has a much larger number of samples. We used SMOTE to create data with a large number of samples, similar to the real environment, and then verified our proposed framework. This amount was approximately 100 times larger than the original dataset. Table 7 shows the time differences between the methods using simple deep learning and the proposed framework based on the oversampled dataset. Because the

number of instances grew enormously, the time required for filtering was considerable. However, the time difference was far more conspicuous than that if we had classified the entire dataset using plain deep learning. When analyzing hundreds of thousands or millions of units of data, the proposed framework is confirmed to significantly reduce the time requirements.

**Table 7.** Changes in time taken for classification according to cost change (unit: Second).

| Cost | J48 Filter + Deep Learning | | | Deep Learning (MLP) |
|------|------|------|------|------|
| | J48 | Deep Learning | Total | |
| 1 | 568.3 | 8307.0 | 8875.3 | |
| 5 | 2071.2 | 8397.2 | 10,468.4 | |
| 10 | 2073.6 | 8310.3 | 10,383.9 | |
| 15 | 2102.7 | 8443.2 | 10,545.9 | 20369.6 |
| 20 | 2093.1 | 8303.8 | 10,396.9 | |
| 25 | 2125.5 | 8302.0 | 10,427.5 | |
| 30 | 2091.7 | 8304.9 | 10,396.6 | |

## 5. Conclusions

In this study, we proposed machine- and deep-learning methods for OTT user analysis to provide ISPs and OTT providers critical timely information about OTT usage data so that they can effectively monitor and execute pricing and mitigation plans. By classifying users according to OTT usage, we confirmed that the classification accuracy was high when using deep learning and conventional machine-learning methods. In particular, deep learning showed higher accuracy. This implies that the application of deep learning to OTT user classification was successful. With plain deep learning, the accuracy of OTT user classification is high, but the classification time takes longer. To shorten this time requirement, we proposed a time-aware MetaCost filtering framework. After first filtering the obvious data using a relatively light algorithm, deep learning was applied to only the most ambiguous data, significantly reducing classification time. However, the accuracy was about the same as with plain deep learning.

This study has the following implications for network and OTT providers. This is the first study that demonstrated how deep learning can be employed to classify OTT user behaviors in a timely manner. ISPs are heavily burdened with applying and maintaining requisite network infrastructure and load balancing to support not only OTT services, but all other internet services, much of which is privately or government contracted. Thus, these investments seriously drive strategy. Hence, timely and extremely accurate usage analysis is needed. This study, therefore, has a wide range of applications in all of those domains.

The proposed framework drastically reduces the time consumption of deep-learning methods with respect to ever-changing user behavior. In fact, when business providers analyze this information, they must consider hundreds of thousands of data items at once. The analysis of such a large amount of data using deep learning can be prohibitively time-consuming and requires heavy computer-resource investments. When applying the proposed method, the costs of time consumption can be drastically reduced.

The proposed method can be used to perform classification according to situations by adjusting the cost factor. In the case where the number of data is relatively small, or there is sufficient time or available resources, accuracy can be improved by increasing the number of data analyzed through deep learning (i.e., cost is set to "high"). On the contrary, if many cases must be analyzed promptly, the cost can be set to "low". Thus, the more obvious data are filtered out. As such, flexible responses are possible by adjusting the cost factor, and the proposed framework can be, therefore, used by providers for real analysis purposes.

In the future, we plan to focus more on the following points. First, when using deep learning, there is a need for a customized methodology suitable for the particular dataset. Because the OTT dataset used in this study comprised unsophisticated features, a simple MLP or CNN resulted in

significant outcomes. However, if complex data were to be analyzed instead, more complex deep learning algorithms must be used. Furthermore, analysis needs to be performed based on various types and categories of OTT user data. To the best of our knowledge, the dataset used in this study is the only public dataset that specializes in OTT. If more datasets related to OTT user behavior will be open to the public in the future, additional and improved research will be possible.

**Author Contributions:** Conceptualization, J.C. and Y.K.; methodology, J.C.; software, J.C.; validation, J.C. and Y.K.; formal analysis, J.C.; investigation, J.C. and Y.K.; resources, J.C.; data curation, J.C.; writing—original draft preparation, J.C.; writing—review and editing, Y.K.; visualization, J.C. and Y.K.; supervision, J.C. and Y.K.; project administration, J.C. and Y.K.; funding acquisition, Y.K. All authors have read and agreed to the published version of the manuscript.

**Funding:** This work was supported by the National Research Foundation of Korea (NRF) grant funded by the Korea government (MSIT) (No. 2020R1G1A1099559).

**Conflicts of Interest:** The authors declare no conflict of interest.

## Appendix A

**Table A1.** Detailed classification accuracy of each filter according to cost change.

| Cost | Filter | KNN + Deep Learning | | | J48 + Deep Learning | | |
|---|---|---|---|---|---|---|---|
| | | Recall | Precision | F-Measure | Recall | Precision | F-Measure |
| 1 | 1st Filter | 0.919 | 0.920 | 0.918 | 0.920 | 0.920 | 0.920 |
| | Deep Learning Filter | 0.947 | 0.960 | 0.951 | 0.953 | 0.962 | 0.956 |
| | Overall | 0.930 | 0.936 | 0.927 | 0.945 | 0.947 | 0.944 |
| 5 | 1st Filter | 0.860 | 0.898 | 0.863 | 0.899 | 0.913 | 0.901 |
| | Deep Learning Filter | 0.944 | 0.947 | 0.945 | 0.944 | 0.948 | 0.945 |
| | Overall | 0.968 | 0.968 | 0.968 | 0.963 | 0.963 | 0.962 |
| 10 | 1st Filter | 0.787 | 0.874 | 0.788 | 0.880 | 0.905 | 0.883 |
| | Deep Learning Filter | 0.944 | 0.947 | 0.945 | 0.948 | 0.950 | 0.949 |
| | Overall | 0.969 | 0.969 | 0.969 | 0.967 | 0.967 | 0.967 |
| 15 | 1st Filter | 0.745 | 0.862 | 0.740 | 0.870 | 0.899 | 0.875 |
| | Deep Learning Filter | 0.950 | 0.951 | 0.951 | 0.949 | 0.951 | 0.950 |
| | Overall | 0.972 | 0.971 | 0.971 | 0.967 | 0.967 | 0.967 |
| 20 | 1st Filter | 0.713 | 0.854 | 0.700 | 0.869 | 0.899 | 0.873 |
| | Deep Learning Filter | 0.955 | 0.956 | 0.955 | 0.951 | 0.952 | 0.951 |
| | Overall | 0.973 | 0.973 | 0.973 | 0.968 | 0.968 | 0.968 |
| 25 | 1st Filter | 0.691 | 0.848 | 0.671 | 0.873 | 0.901 | 0.877 |
| | Deep Learning Filter | 0.959 | 0.959 | 0.959 | 0.951 | 0.952 | 0.951 |
| | Overall | 0.974 | 0.974 | 0.974 | 0.968 | 0.969 | 0.968 |
| 30 | 1st Filter | 0.678 | 0.845 | 0.653 | 0.873 | 0.901 | 0.877 |
| | Deep Learning Filter | 0.963 | 0.963 | 0.963 | 0.953 | 0.956 | 0.954 |
| | Overall | 0.976 | 0.976 | 0.976 | 0.968 | 0.969 | 0.968 |

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
