# Peer review of "Time-Aware Learning Framework for Over-The-Top Consumer Classification Based on Machine- and Deep-Learning Capabilities"

_applsci, doi:10.3390/app10238476_

Round 1

Reviewer 1 Report

Paper Summary.
In this paper, the authors proposed a framework for classifying OTT users by inspecting ISPs' traffic. This method will enable ISPs to determine pricing and mitigation plans according to users' network usage. This framework's unique feature is a two-step classification process that exploits MetaCost filter and the deep-learning methodologies. MetaCost filter classifies the apparent data using a relatively light algorithm and applies a deep-learning filter to only the ambiguous data. The authors have evaluated the accuracy of this approach by utilizing the publicly available OTT traffic data. They have also evaluated the total execution time to classify users by comparing the proposed method with a kNN-based method.

Strengths.
- The paper is well-written, and the topic is relevant to the journal.
- An application scenario looks practical.
- The results are clearly explained, and those results seem very efficient.
- It includes a fulfilling amount of literature reviews.

Weaknesses.
- The DNN structural definition is unclear, and it makes replicability of the proposed method deficient.
- Section 3.1.1 explains already well-known general issues about machine-learning. Also, Section 3.1.2 explains already well-known general issues about DNN. The authors must distinguish the already-known topics and the authors' proposals. Section 3 needs a significant revision.

Author Response

Response to Reviewer 1 Comments

We sincerely thank you for your time and effort in reviewing our paper. Your comments have enabled us to improve the quality and readability of the text. The changes in the revised manuscript file are highlighted in blue and our responses to your comments are included below:

Point 1: The DNN structural definition is unclear, and it makes replicability of the proposed method deficient.

Response 1: We agree with the reviewer on this point. Therefore, we have described how MLP was used in our study in Section 3.1.2 (lines 268–272). In addition, we have also described the rationale behind the structure of CNN used in our research on lines 285–290.

Point 2: Section 3.1.1 explains already well-known general issues about machine-learning. Also, Section 3.1.2 explains already well-known general issues about DNN. The authors must distinguish the already-known topics and the authors' proposals. Section 3 needs a significant revision.

Response 2: Thank you for pointing out this issue.

Our algorithm consists of two parts. In the first part, we verify whether machine learning and deep learning are suitable for analysing OTT users. In the second part, we propose a time-aware framework that utilizes deep learning and MetaCost.

To explain the first part, we included a brief explanation and application method of the machine learning and deep learning methods in Section 3.1. However, this appears to be insufficient. Therefore, we have clarified the composition of this study at the beginning of Section 3 (lines 155–162). The title of Section 3.1 has also been changed for clearer expression. Apart from this, the first line in Section 3.1 and Figure 1 were revised to clarify our research framework.

In addition, in the revisions due to Point 1 above, a detailed deep learning application method was also explained on lines 285-290.

Reviewer 2 Report

I liked reading the paper - it is structured well and there are very few/negligible typos.  The paper addresses an interesting challenge with the motivation and background including the algorithms explained well. Below are some of my comments:

  1. Line 15: amount of time
  2. Line 43: Quality of Service
  3. Line 78: Approximately instead of redundantly?
  4. Line 96: Further?
  5. Line 151: I would suggest that author here discuss the MetaCost method briefly.
  6. Figure 1 can be bigger
  7. Line 196: toward?
  8. Line 287-288: Why? I would expect more discussion on the importance (or not) of such extrema for the ISP/OTT provider.
  9. Line 321: half of the data
  10. Line 348: why?
  11. Line 454: I liked the idea that you used an oversampling method to increase the dataset size but I felt this could have been explained and discussed more. It will certainly improve the contribution of the paper and might also help readers compare their results (with bigger datasets).

Author Response

Response to Reviewer 2 Comments

We sincerely thank you for your time and effort in reviewing our paper. Your comments have enabled us to improve the quality and readability of the text. The changes in the revised manuscript file are highlighted in yellow and our responses to your comments are included below:

Point 1: Line15: amount of time, Line 43: Quality of Service, Line 78: Approximately instead of redundantly?, Line 96: Further?, Line 196: toward?, Line 321: half of the data

Response 1: We have corrected these as follows:

Line 15: amount time → amount of time

Line 43: qualities of service → Quality of Service

Line 78: redundantly → approximately

Line 96: farther → further

Line 202: toward the most data → to most of the data

Line 333: half the data → half of the data

Point 2: Line 151: I would suggest that author here discuss the MetaCost method briefly.

Response 2: As per the suggestion, we have added a brief description of the MetaCost method in lines 151–153.

Point 3: Figure 1 can be bigger.

Response 3: We have increased the size of Figure 1, as per your suggestion.

Point 4: Line 287-288: Why? I would expect more discussion on the importance (or not) of such extrema for the ISP/OTT provider.

Response 4: The characteristics of heavy users or light users can be easily identified. Therefore, we proposed a method to quickly classify them with a simple machine learning technique and classify only the remaining data with deep learning. A supplementary explanation has been added in lines 300–302.

Point 5: Line 348: why?

Response 5: We experimented with adjusting the parameters to select their optimal values for each machine learning technique. The results using the parameters showing the highest performance are included in the paper. A related explanation has been added in lines 360–361

Point 6: Line 454: I liked the idea that you used an oversampling method to increase the dataset size but I felt this could have been explained and discussed more. It will certainly improve the contribution of the paper and might also help readers compare their results (with bigger datasets).

Response 6: We agree with the comment. Therefore, in the revised paper we have explained SMOTE, the oversampling technique we used, and the reason why we used SMOTE. In addition, we have also explained that the proposed algorithm was verified after creating a realistic data set through oversampling (lines 467–473).
